# INFLUENCING HUMANS TO CONFORM TO PREFERENCE MODELS FOR RLHF

## ABSTRACT

Designing a reinforcement learning from human feedback (RLHF) algorithm for learning from preferences requires assuming a preference model, sometimes implicitly. A preference model that poorly describes how humans generate preferences risks learning a poor approximation of the human's unobservable reward function. In this paper, we conduct three human studies to assess whether one can influence the expression of real human preferences to more closely conform to a desired preference model. Importantly, our approach does not seek to alter the human's unobserved reward function. Rather, we change how humans use this reward function to generate preferences, such that they better match whatever preference model is assumed by a particular RLHF algorithm. We introduce three interventions: showing humans the quantities that underlie a preference model, which is normally unobservable information derived from the reward function; training people to follow a specific preference model; and modifying the preference elicitation question. All intervention types show significant effects, providing practical tools to improve preference data quality and the resultant alignment of learned reward functions. Overall we establish a novel research direction in model alignment: training humans and designing interfaces to increase human conformance with the assumptions of the algorithm that will learn from their input.

## 1 INTRODUCTION

Aligning agent behavior with human preferences is a central goal of reinforcement learning from human feedback (RLHF). This process generally assumes a model of human preferences, which defines a probability distribution over a human's rankings of pairs of trajectory segments based on their reward function, which the RLHF algorithm cannot observe.

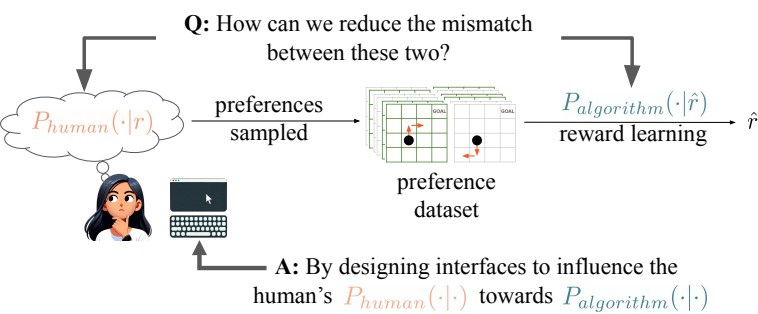

Figure 1: Our proposed method of influencing human preferences.

Prior work has explored different choices for this preference model and provided evidence that the more aligned the RLHF algorithm's preference model is with how humans generate preferences, the more aligned the resulting reward function is (Knox et al., 2022). An unsuitably specified preference model may fundamentally limit the alignment of the learned reward function. Most work assumes that human preferences arise probabilistically from *partial return*, the sum of rewards over a trajectory segment. By this assumption, humans presented with two trajectory segments tend to prefer the one that accrues greater reward, as measured by their reward function (Christiano et al., 2017; Ouyang et al., 2022). Other work has challenged whether partial return is a sufficiently descriptive model of human preferences and proposed alternative models of human preference (Knox et al., 2022; Kim et al., 2023). We instead take a *prescriptive* approach to learning from human prefer-

ences. Specifically, we propose designing training and preference elicitation interfaces to influence humans to better conform to a chosen preference model. We test influencing humans towards the partial return preference model or *regret* preference model, which is based on each segment's deviation from optimality (Knox et al., 2022). See Figure 1 for a visual summary of our interventions.

We introduce three methods and evaluate the ability of each to influence humans towards these two different preference models. First, in the Privileged experiment we present subjects with calculations of regret or partial return for each segment during preference elicitation, thus providing the information needed to exactly follow the *target preference model*. This first intervention is merely a proof of concept, since it requiress knowledge of the reward function. In contrast, the next two methods can be deployed in practice. Second, in the Trained experiment we train humans to follow a specific preference model. Third, in the Question experiment we change *only* the question asked during preference elicitation. The first two methods result in significantly more conformance with either target preference model. The third method also increases conformance with either preference model, but significant effects are only observed when targeting the partial return preference model.

At a high level, we contribute a straightforward approach to improving model alignment: training humans and designing interfaces to increase human conformance with the assumptions of the algorithm that learns from their input. Our experiments yield guidance for RLHF practitioners and lay groundwork for future research at the intersection of interface design and learning from human input, RLHF especially. The code for all computational experiments and interfaces for collecting preferences, as well as the collected datasets of human preferences, are attached with this submission and will be made public upon publication.

## 2 RELATED WORK

### 2.1 LEARNING FROM HUMAN PREFERENCES

Extensive research has explored learning from human preferences for RLHF. This research includes RLHF approaches that explicitly learn a reward function (Christiano et al., 2017; Ibarz et al., 2018; Sadigh et al., 2017; Lee et al., 2021a;b; Ziegler et al., 2019; Ouyang et al., 2022; OpenAI, 2022; Bıyık et al., 2022; Wang et al., 2022; Bai et al., 2022; Glaese et al., 2022; Knox et al., 2022; Touvron et al., 2023; Ethayarajh et al., 2024) and other approaches that directly learn a policy or advantage function from human preferences (Rafailov et al., 2024; Knox et al., 2024; Hejna et al., 2023). All algorithms in the works cited above assume one of the two models of human preference that this paper focuses on. Our investigation of how to increase human conformance with the assumed preference model is compatible with and strengthens this past research.

### 2.2 MODELING HUMAN PREFERENCES

Other research has sought agent alignment by developing preference models that better model human preferences, such as by assuming that human preferences arise from a segment's regret (Knox et al., 2022) or weighted sum of non-Markovian rewards (Kim et al., 2023), instead of a segment's sum of Markovian rewards as is commonly assumed. However, improving the preference model still results in some gap between the preference model and actual human preferences, which are subject to difficult-to-model confounding factors and individual differences. Our research seeks to close this gap for whatever preference model is chosen.

Further, even if we had a *perfect* descriptive model of how *all* humans generate preferences, we may want preferences to be generated by a different preference model that permits more tractable algorithms, greater sample efficiency, or some theoretical guarantees. The methods we introduce can also help in these settings.

## 3 PRELIMINARIES: PREFERENCE MODEL CHOICES

When influencing human preferences, we analyze two preference models: partial return and regret. In this section, we explain the assumptions encoded in each preference model and how each can be used in RLHF.

Consider a Markov decision process (MDP) that represents the task environment using a tuple $(S, A, T, \gamma, D_0, r)$. $S$ and $A$ are the sets of possible states and actions, respectively. $T : S \times A \to p(\cdot|s, a)$ is a transition function; $\gamma$ is the discount factor; and $D_0$ is the distribution of start states. Unless stated otherwise, we assume tasks are undiscounted ($\gamma = 1$) and have terminal states, after which only 0 reward can be received. $r$ is a reward function, $r : S \times A \times S \to \mathbb{R}$, where $r_t$ is a function of $s_t$, $a_t$, and $s_{t+1}$ at time $t$. An MDP\$r$ is an MDP without a reward function.

Let $r$ refer to the ground-truth reward function for some MDP, $\hat{r}$ refer to a learned approximation of $r$, and $\tilde{r}$ refer to any reward function (including $r$ or $\hat{r}$). A policy ($\pi : S \times A \to [0, 1]$) specifies the probability of an action given a state. $Q_{\tilde{r}}^\pi$ and $V_{\tilde{r}}^\pi$ refer respectively to the state-action value function and state value function for a policy, $\pi$, under $\tilde{r}$, and are defined as follows: $V_{\tilde{r}}^\pi(s) \triangleq \mathbb{E}_\pi[\sum_{t=0}^{\infty} \tilde{r}(s_t, a_t, s_{t+1})|s_0 = s]$ and $Q_{\tilde{r}}^\pi(s, a) \triangleq \mathbb{E}_\pi[\tilde{r}(s, a, s') + V_{\tilde{r}}^\pi(s')]$.

An optimal policy $\pi_{\tilde{r}}^*$ is any policy where $V_{\tilde{r}}^{\pi_{\tilde{r}}^*}(s) \geq V_{\tilde{r}}^\pi(s)$ at every state $s$ for every policy $\pi$. We write shorthand for $Q_{\tilde{r}}^{\pi_{\tilde{r}}^*}$ and $V_{\tilde{r}}^{\pi_{\tilde{r}}^*}$ as $Q_{\tilde{r}}^*$ and $V_{\tilde{r}}^*$, respectively.

### 3.1 Reward Learning from Pairwise Preferences

RLHF typically learns a reward function by minimizing the cross-entropy loss—i.e., maximizing the likelihood—of observed human preference labels (Christiano et al., 2017; Ibarz et al., 2018; Wang et al., 2022; Bıyık et al., 2021; Sadigh et al., 2017; Lee et al., 2021a;b; Ziegler et al., 2019; Ouyang et al., 2022; Bai et al., 2022; Glaese et al., 2022; OpenAI, 2022; Touvron et al., 2023; Hejna III & Sadigh, 2023).

**Segments** Let $\sigma$ denote a segment starting at state $s_0^\sigma$. Its length $|\sigma|$ is the number of transitions within the segment. A segment includes $|\sigma| + 1$ states and $|\sigma|$ actions: $(s_0^\sigma, a_0^\sigma, s_1^\sigma, a_1^\sigma, ..., s_{|\sigma|}^\sigma)$. In this problem setting, segments lack any reward information. As shorthand, we define $\sigma_t \triangleq (s_t^\sigma, a_t^\sigma, s_{t+1}^\sigma)$. Additionally, we denote the *partial return* of a segment $\sigma$ as $\Sigma_\sigma \tilde{r}$ for some $\tilde{r}$, where $\tilde{r}_t^\sigma \triangleq \tilde{r}(s_t^\sigma, a_t^\sigma, s_{t+1}^\sigma)$ and $\Sigma_\sigma \tilde{r} \triangleq \sum_{t=0}^{|\sigma|-1} \tilde{r}_t^\sigma$.

**Preference datasets** We denote a preference dataset as $D_\succ$, which comprises samples of preferences over pairs of segments. Each sample is represented as $(\sigma_1, \sigma_2, \mu)$. Vector $\mu = \langle \mu_1, \mu_2 \rangle$ represents the preference; specifically, if $\sigma_1$ is preferred over $\sigma_2$, denoted $\sigma_1 \succ \sigma_2$, $\mu = \langle 1, 0 \rangle$. $\mu$ is $\langle 0, 1 \rangle$ if $\sigma_1 \prec \sigma_2$ and is $\langle 0.5, 0.5 \rangle$ for $\sigma_1 \sim \sigma_2$ (no preference). For a sample $(\sigma_1, \sigma_2, \mu)$, we assume that the two segments have equal lengths (i.e., $|\sigma_1| = |\sigma_2|$) and the same start state (i.e., $s_0^{\sigma_1} = s_0^{\sigma_2}$).

**Loss function** When learning a reward function from a preference dataset, $D_\succ$, preference labels are typically assumed to be generated by a preference model $P$ based on an unobservable *ground-truth* reward function $r$. We learn $\hat{r}$, an approximation of $r$, by minimizing this cross-entropy loss:

$$loss(\hat{r}, D_\succ) = -\sum_{(\sigma_1, \sigma_2, \mu) \in D_\succ} \mu_1 \log P(\sigma_1 \succ \sigma_2 | \hat{r}) + \mu_2 \log P(\sigma_1 \prec \sigma_2 | \hat{r}) \tag{1}$$

If $\sigma_1 \succ \sigma_2$, the sample's likelihood is $P(\sigma_1 \succ \sigma_2 | \hat{r})$ and its loss is therefore $-log P(\sigma_1 \succ \sigma_2 | \hat{r})$. If $\sigma_1 \prec \sigma_2$, its likelihood is $1 - P(\sigma_1 \succ \sigma_2 | \hat{r})$. This loss is under-specified until the preference model $P(\sigma_1 \succ \sigma_2 | \hat{r})$ is defined. Learning approximations of $r$ from preferences can be summarized as "minimize Equation 1".

**Preference models** A preference model determines the likelihood of one trajectory segment being preferred over another, $P(\sigma_1 \succ \sigma_2 | \tilde{r})$.

### 3.2 Preference Models: Partial Return and Regret

Here we describe the two preference models that are most commonly used when learning from human preferences. Figure 2 illustrates an example of how these two models differ. In Section 5 we detail our proposed training procedures and preference elicitation interfaces that influence human preferences to conform to either choice of preference model. We focus on two preference models that differ solely in the segment statistic they use to evaluate the desirability of a trajectory segment. We do not address other aspects of modeling human preferences, such as different assumptions about human irrationality, risk aversion, or uncertainty. However, the interventions proposed in this

paper, which aim to influence humans towards a chosen preference model, may be generalizable to influencing humans to conform to preference models that incorporate these additional factors.

**Partial return** The most common preference model (e.g., Christiano et al. (2017)) posits that human preferences are generated by a Boltzmann distribution over the partial returns of the two segments, expressed here as a logistic function:

$$P_{\Sigma_r}(\sigma_1 \succ \sigma_2|\tilde{r}) = logistic\Big(\Sigma_{\sigma_1}\tilde{r} - \Sigma_{\sigma_2}\tilde{r}\Big). \quad (2)$$

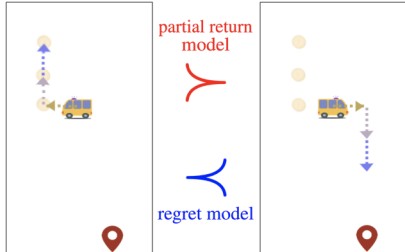

**Regret** An alternative model of human preferences is based on regret (Knox et al., 2022). This model suggests that human preferences arise from the deviations of each segment from optimal decision-making, characterized by the regret of each transition within the segment. In this paper, we only focus on segments with deterministic transitions, although the regret preference model equally accommodates segments with stochastic transitions. For a single deterministic transition $(s_t, a_t, s_{t+1})$, $regret_d(\sigma_t|\tilde{r}) \triangleq V_{\tilde{r}}^*(s_t^\sigma) - [\tilde{r}_t + V_{\tilde{r}}^*(s_{t+1}^\sigma)]$. For a full segment,

Figure 2: On each step, the yellow vehicle receives reward as the sum of reward components: $-1$ for every time it moves; $+1$ for collecting a coin; and $+50$ for reaching the red goal marker, which ends the episode. The partial return preference model favors the left trajectory, while regret favors the right.

$$regret_d(\sigma|\tilde{r}) \triangleq \sum_{t=0}^{|\sigma|-1} regret_d(\sigma_t|\tilde{r})$$
$$= V_{\tilde{r}}^*(s_0^\sigma) - (\Sigma_\sigma \tilde{r} + V_{\tilde{r}}^*(s_{|\sigma|}^\sigma)), \quad (3)$$

with the right-hand expression arising from cancelling out intermediate state values. Equation 3 only applies to deterministic transitions; Knox et al. (2022) provide the general formulation of regret. The default version of the regret preference model is the Boltzmann distribution over the *negated* regret:

$$P_{regret}(\sigma_1 \succ \sigma_2|\tilde{r}) = logistic\Big(regret_d(\sigma_2|\tilde{r}) - regret_d(\sigma_1|\tilde{r})\Big).$$

See Appendix B for further explanation of regret, of partial return, and of how these preference models differ. We refer to the regret and partial return of a segment as types of *segment statistics*.

Knox et al. (2022) demonstrated that the regret preference model better fits a dataset of human preferences, leading to more human-aligned learned reward functions. Knox et al. (2024) showed that the predominant method for fine-tuning large language models by RLHF can be derived from either preference model, but the derivation from the regret preference model avoids arbitrarily setting the discount factor $\gamma = 0$. From the regret preference model, Hejna et al. (2023) derive constrastive preference learning (CPL), which learns a policy *directly* from human preferences and thereby avoids the challenge during learning of modeling optimal value functions for each time it is updated.

Nonetheless, using the partial return preference model remains appealing, given the greater simplicity of learning a reward function with it and the amount of research that has been built upon it. Therefore our goal is to assess whether it is possible to influence humans to adopt a given preference model, whether regret or partial return.

In the descriptions above of the two preference models, both are defined narrowly to assume that humans follow the Boltzmann distribution. This assumption is ubiquitous in RLHF but has limitations, which Zhu et al. (2024) overview. Both partial return and regret can, however, inform probability distributions other than the Bolzmann distribution. Therefore, in our evaluations (Section 5) we additionally test whether humans conform to the *noiseless* version of a preference model, which deterministically prefers the segment with the more desirable statistic (e.g., higher partial return for the noiseless partial return preference model or lower regret for the noiseless regret preference model).

## 4 EXPERIMENTAL TASK AND PREFERENCE ELICITATION PROCEDURE

To empirically investigate our methodology for influencing human preferences, we collected preference datasets labeled by human subjects with IRB approval. This section provides an overview of

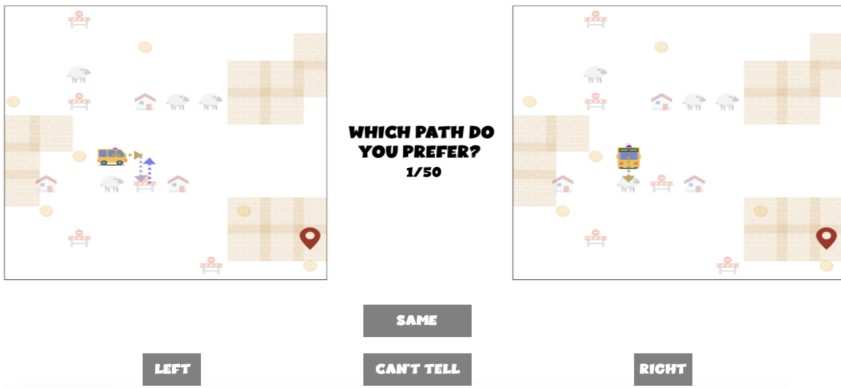

Figure 4: The baseline preference elicitation interface shown to humans annotators. Two of our three experiments—the Privileged experiment and the Question experiment—-involve changes to this interface.

the user interface elements shared by each experiment introduced in Section 5. See Appendix C for further details.

**Subject training**   Subjects first learn about the general grid-world delivery domain, for which task instantiations are created by a map of objects and terrain. The subjects specifically are taught about the reward function and how objects and terrain affect the agent's transitions. As part of this teaching, subjects control the agent on domain maps designed to teach one or two concepts at a time.

**Preference elicitation interface**   After teaching subjects to understand the domain and delivery task, we elicit their preferences. Figure 4 illustrates a baseline version of the preference elicitation interface. In this work, we exclude preferences labeled "can't tell." After preference elicitation, a survey tests the subject's task comprehension and attentiveness, as detailed in Appendix C.4.

Figure 3: The delivery task shown to human subjects for gathering preferences. The yellow vehicle is the agent, and its objective is to maximize its score. Score maximization requires reaching the red inverted teardrop.

# 5   EXPERIMENTAL EVALUATION
## OF THREE METHODS OF INFLUENCE

Aiming to decrease the gap between an RLHF algorithm's assumed preference model and how a preference dataset is actually generated by humans, we conduct three random-assignment experiments for collecting human preferences. Each experiment represents a type of intervention and has three conditions that each result in a preference dataset: a control condition, an intervention condition that influences subjects to follow the regret preference model ($P_{regret}$), and an intervention condition that influences subjects to follow the partial return preference model ($P_{\Sigma_r}$). Thus, the ability of each type of intervention to influence the human towards a preference model is tested with two different preference models.

- **Privileged experiment** (Section 5.1) - The intervention of this experiment is to display information about each segment's regret or partial return under the ground-truth reward function during preference elicitation. We refer to this information as privileged because it relies upon the ground-truth reward function, which in practical settings is unknown by the code underlying the preference elicitation interface.

- **Trained experiment** (Section 5.2) - The intervention of this experiment is to train subjects to give preferences based upon either partial return or regret. Training includes how to calculate a segment's regret or partial return.

- **Question experiment** (Section 5.3) - The intervention of this experiment changes the question that is asked during preference elicitation (see Figure 4) to one designed to encourage adherence to one of the preference models.

In all experiments, the ground-truth reward function remains the same and is taught to subjects before preference elicitation to enable our analysis, but it is unavailable to the reward learning algorithm, which relies solely on the generated preference dataset. Only the Privileged experiment leverages the ground-truth reward function in its preference elicitation interface design.

## 5.1 PRIVILEGED EXPERIMENT

We first study whether providing subjects with privileged information about a preference model during preference elicitation influences their preferences towards that model. Presenting this privileged information serves as a probe into how susceptible to influence human preferences are.

When presented with two segments for preference labeling, subjects in all three conditions are asked "Which shows better behavior?". This question was later refined for the other experiments to the baseline question in Figure 4. Each subject labeled preferences between 35 to 50 segment pairs. After data filtering (see Appendix C), our datasets come from from 64 subjects in the $P_{\Sigma_r}$-Privileged condition (video walk-through of the interface), 65 subjects in the $P_{regret}$-Privileged condition (video walk-through), and 50 subjects in the Privileged-Control condition (video walk-through). We refer to each condition's resultant preference dataset by the condition's name.

**Intervention details: subject training** In the $P_{\Sigma_r}$-Privileged condition, subjects are shown each segment's partial return during preference elicitation. Likewise, subjects in the $P_{regret}$-Privileged condition are shown each segment's regret. Examples can be found in Figure 16 of the Appendix. Note that we do not explicitly instruct subjects to use the displayed segment statistics when labeling preferences. Rather, it is merely made visible. In the third condition, Privileged-Control, subjects are not shown any information during preference elicitation other than the visualization of the two segments that all subjects in these experiments see. See Appendix D for further details. Conditions differ only as discussed above.

**Hypothesis 1:** Presenting each segment's statistic—whether partial return or regret—will influence the human to give preferences according to this statistic.

To test this hypothesis, we compare how well the target preference model—i.e., the preference model the intervention aims to influence human preferences towards—predicts the resultant dataset of preferences for the corresponding condition and for the control condition. Specifically, the preference model is given the ground-truth reward function, scaled by a constant positive factor. Given a target preference model, we compare the mean cross entropy (i.e., negative log likelihood) of each condition's preference dataset. Noting that all positive scalings of the reward function order policies equivalently and also only affect the Boltzmann distribution as a temperature parameter would, we choose the scaling constant that results in the lowest cross entropy during a grid search.

The cross entropy losses plotted in Figure 5 support Hypothesis 1. In particular, for both intervention conditions, presenting privileged information about the target preference model results in a dataset with a higher likelihood than that of the Privileged-Control

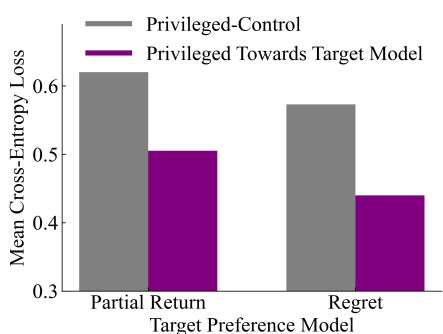

Figure 5: For the Privileged experiment, mean cross-entropy loss over each condition's preference dataset with respect to the target preference model. If the loss is lower for an intervention's dataset than for the Privileged-Control dataset, then the former is better predicted by the target preference model. Performing a Mann-Whitney U test results in $p < 0.01$ for both conditions.

dataset. Because samples in each condition are unpaired, we perform a Mann-Whitney U test comparing the likelihood of samples in the Privileged-Control dataset to those in the intervention condition's dataset. Both interventions result in a statistically significant effect ($p < 0.01$). Appendix H provides plots of the cross-entropy loss for all reward scaling parameters for each experiment,

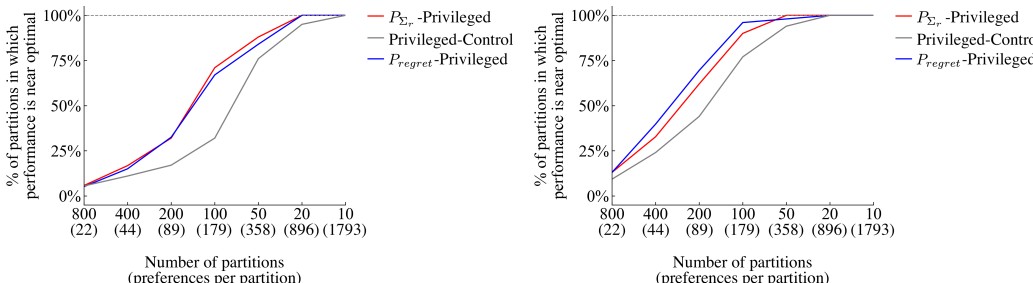

Figure 6: Learning a reward function with the partial return preference model (Left) and regret preference model (Right) from the preferences collected in the Privileged experiment. Each preference dataset is randomly partitioned into equal-sized training sets 10 times, using 10 different random seeds to control partitioning. We learn a reward function using the given preference model for each training set, and plot the percent of partitions—or equivalently training sets—in which the learned reward function induces near-optimal behavior. Note this percentage is across all partitions across all 10 seeds. To visually test for Hypothesis 2, observe the gap between the colored (i.e., red and blue) lines and the gray line.

and more details on the statistical test. Additionally, Appendix H provides an explanation of how we disentangle the effects of learning more about the ground-truth reward function versus the target preference model, addressing concerns that the proposed interventions may merely teach human subjects more about the ground-truth reward function rather than the target preference model.

We remove the assumption that humans give preferences according to the Boltzmann distribution—used in the above likelihood test—by computing the accuracy for a noiseless version of each preference model. A Fisher's exact test also finds significant effects for both interventions ($p < 0.01$), supporting Hypothesis 1. See Appendix I for details about our accuracy testing.

**Hypothesis 2:** Presenting each segment's statistic leads to learning more aligned reward functions with the preference model corresponding to this statistic.

For each condition's dataset, we learn a reward function $\hat{r}$ by minimizing Equation 1 when assuming $P_{regret}$ or $P_{\Sigma_r}$. In all experiments, regret is approximated via the algorithm proposed by Knox et al. (2022), employing successor features to efficiently compute differentiable estimates of the value functions in Equation 3. Value iteration (Sutton & Barto, 2018) then computes the approximately optimal $Q_{\hat{r}}^*$ function, and we then derive the maximum-entropy optimal policy from $Q_{\hat{r}}^*$, which uniformly randomly selects among all optimal actions. We assess the mean return of the resulting policy relative to the ground-truth reward function $r$ over the initial state distribution $D_0$.

We normalize the mean return of a policy $\pi$, $V_r^\pi$, using the formula $(V_r^\pi - V_r^U)/V_r^*$. Note that $V_r^*$ denotes the expected return of an optimal policy and $V_r^U$ denotes the expected return of a uniformly random policy, both computed over $D_0$. A normalized mean return greater than 0 is better than $V_r^U$, while a value of 1 corresponds to optimal performance. We consider a normalized return above 0.9 to signify near-optimal performance.

For each dataset, we randomly assign human preferences to different numbers of same-sized partitions. Each partition is used as a training set, and we evaluate with 10 different random seeds that control how preferences are partitioned. Results are shown in Figure 6, with reward learning details outlined in Appendix J for all experiments. Appendix J additionally presents results comparing performance of the resultant policy to that of a uniformly random policy ($V_r^U$).

Figure 6 supports Hypothesis 2. Learning with the partial return preference model from the $P_{\Sigma_r}$-Privileged dataset results in reward functions that induce near-optimal behavior more often than when learning from the Privileged-Control dataset for all partition sizes, except for the largest partition size where performance is matched. A comparable pattern is observed when learning from the $P_{regret}$-Privileged dataset with the regret preference model.

## 5.2 TRAINED EXPERIMENT

In real-world practice, RLHF practitioners do not have access to the ground-truth reward function during preference elicitation. Therefore, in this experiment and the next, we focus on interventions to improve preference model alignment that are feasible in practice. In the TRAINED EXPERIMENT,

we evaluate training humans to follow either the partial return or regret preference models during preference elicitation. All experimental details are the same as those of the PRIVILEGED experiment unless otherwise noted.

This experiment consists of three conditions. In the $P_{\Sigma_r}$-Trained and $P_{regret}$-Trained intervention conditions, subjects are taught to follow the corresponding preference model. In the Trained-Control condition, subjects are not taught about any specific preference model and are trained identically as all subjects in the PRIVILEGED experiment. The conditions differ only by how subjects are trained. Subjects are randomly assigned to a condition until each condition has data from 10 subjects. Unlike in the PRIVILEGED experiment, if a subject's preference data is removed because of poor task comprehension or inattentiveness (see Appendix C.4), their slot is reopened for random assignment, including their exact set of segment pairs to be labeled with preferences. With this replacement strategy, we gather preferences between the same set of segment pairs for each condition, which produces paired data. We discuss this design choice in Appendix H. A video walk-through of the interface used for the $P_{\Sigma_r}$-Trained condition is available here, for the $P_{regret}$-Trained condition here, and for the Trained-Control condition here.

**Intervention details: subject training**    During training, subjects in the intervention conditions learn about the domain's dynamics and reward function, as well as the segment statistic specific to their condition (partial return or regret). They are shown the segment statistic while interacting with the delivery domain and taught how to compute it. We then provide subjects with a detailed example of how to use the taught segment statistic to generate preferences, have them practice with feedback, and finally ask them to label preferences over segment pairs from the delivery task. Subjects label preferences for 50 segment pairs with the elicitation question changed based on the condition. Further details on the training protocol are provided in Appendix E.

**Hypothesis 1:** Training a human to follow a specific preference model will influence the human to give preferences according to that model.

We follow the procedure outlined in Section 5.1 to evaluate evaluate Hypothesis 1 with results shown in Figure 7. Hypothesis 1 is supported; the intervention condition datasets are more likely under the respective target preference model than the control condition. Because samples are paired, we perform a Wilcoxon paired signed-rank test comparing the likelihood of samples in the control dataset to those in the dataset from training humans to follow a preference model, finding a statistically significant difference at $p < 0.01$ that further supports Hypothesis 1. See Appendix H for details. Additionally, we find that the accuracy over both the $P_{\Sigma_r}$-Trained and $P_{regret}$-Trained datasets is notably higher than the accuracy over the Trained-Control dataset given the noiseless version of the respective target preference

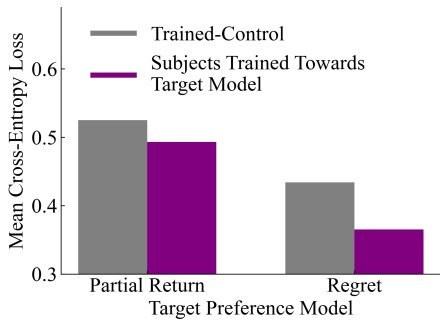

Figure 7: For the Trained experiment, mean cross-entropy loss over each condition's preference dataset with respect to the target preference model. Lower is better. Performing a Wilcoxon paired signed-rank test results in $p < 0.01$ for both conditions.

model. These results, detailed in Appendix I, additionally support Hypothesis 1 with statistical significance at $p < 0.01$.

**Hypothesis 2:** Training humans to follow a specific preference model leads to learning more aligned reward functions with that preference model.

Figure 8 illustrates partial support for Hypothesis 2. Learning from the $P_{regret}$-Trained dataset results in reward functions that induce near-optimal performance more often than learning from the Trained-Control dataset when using the regret preference model for learning. But, learning from the $P_{\Sigma_r}$-Trained dataset when using the partial return preference model leads to comparatively poor performance. See Appendix J.6 for a discussion on this result.

For larger training sets, learning a reward function with the partial return preference model using the $P_{regret}$-Trained dataset instead of the other condition's datasets results in reward functions that induce near-optimal behavior more often; influencing preferences towards $P_{regret}$ appears to be generally beneficial for reward learning in this task.

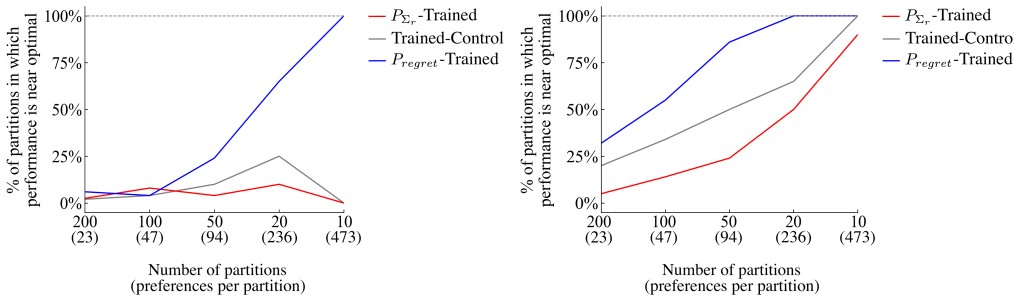

Figure 8: Learning a reward function with the partial return preference model (Left) and regret preference model (Right) from the preferences collected in the Trained experiment. See Figure 6 for more details on how this figure was generated.

Additionally, in Appendix E, we provide results indicating that subjects generally agreed with how the taught preference model generated preferences, but find no statistically significant correlation between subjects' ability to compute the relevant segment statistics and their adherence to the target preference model during preference elicitation.

## 5.3 QUESTION EXPERIMENT

For some domains, it may be difficult or time consuming to teach humans about a specific preference model. Concepts such as the partial return of a trajectory segment may be difficult for humans to comprehend in other environments, for example when eliciting preferences over large language model outputs. Instead, we change only the wording of the question asked during preference elicitation, investigating how to align human preferences with specific preference model without relying on their explicit understanding of the preference model.

This experiment consists of three conditions; the $P_{\Sigma_r}$-Question and $P_{regret}$-Question conditions, where we change the preference elicitation instruction in favor of the partial return preference model and regret preference model, respectively, and the Question-Control condition where we use a preference elicitation question that does not seek to influence human preferences towards any specific preference model. Subjects' training matches that of the previous experiment's control condition, and each condition differs only in the wording of the question we ask when eliciting preferences from subjects. We collect data from 9 subjects per condition who are assigned to conditions via random assignment. We replace the data of subjects that are removed due to poor task comprehension or inattentiveness. A video walk-through of the interface used for the $P_{\Sigma_r}$-Question condition is available here, for the $P_{regret}$-Question condition here, and for the Question-Control condition here. Appendix F contains further details.

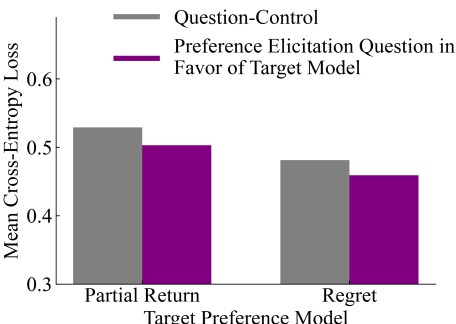

Figure 9: For the Question experiment, mean cross-entropy loss over each condition's preference dataset with respect to the target preference model. Lower is better. Performing a Wilcoxon paired signed-rank test results in significance for only the $P_{regret}$-Question condition ($p < 0.01$).

**Intervention details: Preference elicitation questions**    The subjects in each condition see the corresponding question below:

- Question-Control - "Which path do you prefer?", chosen to reduce influence

- $P_{\Sigma_r}$-Question - "Which path has better immediate outcomes?", chosen to focus subjects only on the reward within a segment

- $P_{regret}$-Question - "Which path reflects better decision-making?", chosen to reflect regret's measurement of a segment's deviation from optimality

**Hypothesis 1:** Changing the preference elicitation question in favor of a specific preference model will influence the human to give preferences according to that model.

Results are shown in Figure 9, supporting Hypothesis 1 with a relatively small effect size. The loss for the $P_{\Sigma_r}$-Question dataset under the partial return preference model is lower than the loss for the Question-Control dataset, indicating a small shift in subject's preferences towards the partial return preference model. A Wilcoxon paired signed-rank test between the likelihoods of these two datasets indicates a statistically significant difference at $p < 0.05$. The $P_{regret}$-Question dataset's loss under the regret preference model is also lower than, but close to, that of the Question-Control's dataset with no statistically significant difference. We observe a similar pattern when looking at the accuracy of the noiseless target preference model over the $P_{\Sigma_r}$-Question and $P_{regret}$-Question datasets with similar significance. See Appendix H and I for more details.

**Hypothesis 2:** Changing the preference elicitation question in favor of a specific preference model leads to learning more aligned reward functions with that preference model.

Figure 10 provides evidence in support of Hypothesis 2. Modifying the preference elicitation question to steer human preferences towards a specific preference model results in a dataset that induces near-optimal behavior more often—when learned from using the target preference model—compared to the control condition dataset. Broadly speaking, this suggests that the question we ask subjects when labeling preferences can effect the performance of the resulting learned reward function.

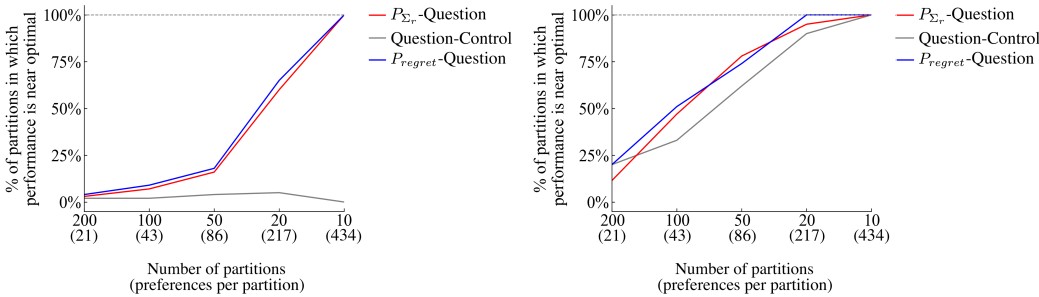

Figure 10: Learning a reward function with the partial return preference model (Left) and regret preference model (Right) from the preferences collected in the Question experiment. See Figure 6 for more details on how this figure was generated.

## 6 CONCLUSION

The choice of preference model used by an RLHF algorithm introduces a source of misalignment between how humans are assumed to generate preferences and how they actually generate preferences, potentially limiting the alignment of the learned reward function. Even if we could *perfectly* model *all* human preferences, we may wish that preferences are generated by a different model that is computationally efficient for RLHF or provides certain theoretical guarantees. To this end, we propose influencing human preferences towards a chosen preference model through user-interface design—a novel direction for RLHF research.

We first establish that humans can be significantly influenced towards a specific preference model when privileged information about that model is shown during preference elicitation. We then introduce two practical interventions; training subjects to follow a preference model in the Trained experiment—which significantly influences them towards that specific model—-and changing the preference elicitation question in the Question experiment—which can significantly influence humans towards the partial return preference model and moderately towards the regret preference model. All three interventions result in learning more aligned reward functions.

Our findings suggest that human training and preference elicitation interfaces should be viewed as essential tools for improving alignment in RLHF. Appendix A highlights the potential practicality of our approach; the Trained and Question interventions demonstrate important potential for real-world application. Notably, the Question experiment offers a viable path forward by influencing human preferences towards a specific preference model without requiring knowledge of the ground-truth reward function, while the Trained experiment establishes a foundation for extending this methodology to real-world domains.

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
