# OpenReview forum: "Influencing Humans to Conform to Preference Models for RLHF"
_ICLR.cc/2025/Conference — Submitted to ICLR 2025_

### Official Review · Reviewer_6yvA · 2024-10-28

**Soundness:** 3
**Presentation:** 4
**Contribution:** 2
**Rating:** 5
**Confidence:** 4

**Summary:**

The paper investigates methods to align human preferences with a specified preference model in RLHF by influencing how humans express preferences. This approach doesn't alter the underlying human reward function but instead modifies how preferences are generated to match an RLHF algorithm’s assumed model. Three interventions were tested across human studies: Privileged Experiment, Trained Experiment, and Question Experiment. The paper claims that each method effectively improved alignment with the target model.

**Strengths:**

- The paper is generally well-written and clear.
- The experimental setup is described in detail, providing clarity on the methodology.
- Thorough analysis of the experiments strengthens the overall findings.

**Weaknesses:**

- RLHF’s value in aligning language models is widely recognized, but the paper’s focus on a constrained grid-world delivery domain may limit generalization to language-based agents.
    * While the proposed interventions show promise, it’s unclear how effectively these ideas could extend to more complex language-driven settings.
- The environment and experiment setting closely follow prior work by Knox et al., meaning the primary contribution arises from the three interventions aimed at influencing human decision-making: the Privileged Experiment, Trained Experiment, and Question Experiment. However, concerns remain about both the novelty and practicality of these experiments:
    * The Trained Experiment, which involves training participants before they label data, is resource-intensive, with significant demands on time and cost, as noted by the authors.
    * The Question Experiment has only a marginal effect, with no significant impact observed for aligning preferences with the regret model, which may limit its utility in practice.

**Questions:**

- Does influencing preferences toward a specific model potentially introduce bias in scenarios where the model may not be the best fit for human objectives?
- In the Privileged Experiment, given that users see reward information directly, is there a risk they might over-rely on this data, reducing their engagement with the task?
- Could the interventions used in the study generalize to more complex RLHF tasks (e.g., continuous environments) where users make dynamic, long-term decisions?

---

> ### Author Response · Authors · 2024-11-16
>
> Thank you for your review! We hope we have addressed your concerns regarding the practicality of our proposed approach and its potential extension to more complex domains in our response to all reviewers. Applying our approach to language-driven settings is an interesting direction for future exploration. To answer your other questions:
>
> “Does influencing preferences toward a specific model potentially introduce bias in scenarios where the model may not be the best fit for human objectives?”
>
> We assume there exists some preference model that an RLHF practitioner wishes to use during reward learning, and outline several interventions to influence the expression of human preferences towards this preference model. What that specific preference model should be, including if it is an adequate fit to learn the human objectives, remains at the discretion of the RLHF practitioner. Note that prior work in RLHF typically assumes a preference model; we propose providing the additional flexibility of influencing humans towards this preference model.
>
> “In the Privileged Experiment, given that users see reward information directly, is there a risk they might over-rely on this data, reducing their engagement with the task?”
>
> This is an important insight, however as noted in the paper (see Section 5.1), the Privileged experiment serves as a probe into whether or not it is possible to influence the expression of real human preferences. The Trained and Question experiments aim to provide a practical methodology for influencing human preferences towards a specific preference model, where this is no longer a concern.

---

> > ### Comment · Reviewer_6yvA · 2024-11-28
> >
> > Thank you for your response. While the topic and insights are interesting, I remain concerned about the practicality of the experiments. The trained experiment is highly resource-intensive, and the question experiment demonstrates only a marginal effect. As such, I will maintain my current score.

---

### Official Review · Reviewer_gCy4 · 2024-10-31

**Soundness:** 2
**Presentation:** 2
**Contribution:** 2
**Rating:** 5
**Confidence:** 2

**Summary:**

This paper explores methods to improve the quality of human feedback in reinforcement learning from human feedback (RLHF) by aligning human preference expressions with the preference model assumed by the RLHF algorithm. Poorly aligned preferences can lead to inaccurate approximations of a human's reward function. Through three human studies, the authors assess interventions to influence how humans express preferences without altering their underlying reward function. These interventions include displaying model-based preference quantities, training people to follow specific models, and rephrasing preference questions. All three interventions significantly improve preference alignment, introducing a novel direction in model alignment by adjusting human interaction to better match algorithmic assumptions.

**Strengths:**

Originality: Very original work - First one I've seen on aligning human preference expressions with the assumptions of the preference model, an interesting and out-of-the-box perspective.

Quality: Good variety of experiments to test the hypotheses, covering multiple different ways to influence human preference expressions.

Clarity: Very clear introduction, related work, description of tasks, and description of experimental interventions

Significance: Interesting and novel research direction which could lead to significant improvements in RLHF

**Weaknesses:**

Typos:
- Figure 5 caption: "the the"
- Line 381: "Priveleged"

I feel like the "% of partitions in which performance is near optimal" results are not very convincing to support the hypotheses presented.

PRIVILEGED EXPERIMENT
Figure 6: The difference between the P_regret and P_reward lines in both subplots is very minimal and does not hold for all number of partitions.

TRAINED EXPERIMENT
Figure 8: The P_regret trained preferences always lead to better performance -- this seems to indicate that the P_regret training paradigm leads to better understanding of the ground-truth reward function more than it supports Hypothesis 2.

QUESTION EXPERIMENT
The P_regret question does not seem to me to focus subjects on a segment's deviation from optimality. Perhaps something like "Which path is farthest from optimal?" would have been better
Figure 10: The difference between the P_regret and P_reward lines in both subplots is very minimal and does not hold for all number of partitions.

**Questions:**

Please address the weaknesses listed above.

---

> ### Author Response · Authors · 2024-11-16
>
> Thank you for your review and for recognizing the novelty and significance of our paper. We also appreciate your careful attention to detail in pointing out several typos. Regarding the potential weaknesses you mentioned:
>
> “I feel like the "% of partitions in which performance is near optimal" results are not very convincing to support the hypotheses presented.”
>
> Our main objective, as expressed in Hypothesis 1 for each experiment, is to demonstrate that each intervention can influence humans toward a specific preference model, and we generally succeed in showing this. The figures you reference (Figures 6, 8, and 10) pertain to Hypothesis 2, which focuses on reward learning from a dataset of preferences influenced towards a specific preference model. With this distinction in mind, we will now further address your concerns.
>
> “PRIVILEGED EXPERIMENT Figure 6: The difference between the P_regret and P_reward lines in both subplots is very minimal and does not hold for all number of partitions.”
>
> We agree with your observation. However, to evaluate Hypothesis 2, it is important to focus on the gap between the blue/red lines and the gray line in Figure 6. The blue/red lines represent learning from preference datasets where humans were influenced toward the regret/partial return models, while the gray line reflects learning from the control condition’s dataset. A significant gap exists for all partition sizes except the smallest and largest, suggesting that presenting each segment’s statistic (blue/red lines) leads to learning more aligned reward functions with the preference model corresponding to this statistic than when learning from the control condition’s dataset (gray line). Comparison between the blue lines and the red lines is irrelevant to Hypothesis 2. Therefore, Figure 6 supports Hypothesis 2.
>
> “TRAINED EXPERIMENT Figure 8: The P_regret trained preferences always lead to better performance -- this seems to indicate that the P_regret training paradigm leads to better understanding of the ground-truth reward function more than it supports Hypothesis 2.”
>
> We agree with your observation that teaching humans to follow the regret preference model always leads to better performance when learning a reward function, but disagree with your proposed explanation that this is because the Trained intervention simply leads to a better understanding of the ground-truth reward function. We address this concern in the response posted to all reviewers. The results in Figure 8 are actually in line with prior work, which found that learning from preferences generated by the regret preference model induces more aligned reward functions (see Knox et al., 2022)  and that preferences generated by the regret preference model—when learned from under the assumption that instead they were generated by the partial return preference model—results in a highly shaped reward function in a variety of conditions (see Knox et al., 2024). Therefore, influencing humans towards the regret preference model reasonably leads to better reward learning performance.
>
> “QUESTION EXPERIMENT The P_regret question does not seem to me to focus subjects on a segment's deviation from optimality. Perhaps something like "Which path is farthest from optimal?" would have been better Figure 10: The difference between the P_regret and P_reward lines in both subplots is very minimal and does not hold for all number of partitions.”
>
> We appreciate your careful attention to the wording used to influence humans toward the regret preference model. Since we elicited preferences from subjects without assuming technical backgrounds, we deliberately avoided terms like 'optimal,' which carry specific meanings in computer science but may not be as clear to the broader public. Regarding your observation about Figure 10, to evaluate Hypothesis 2, we focus on comparing the blue/red lines with the gray line as we described above for Figure 6. Specifically, we are interested in the effects of influencing humans toward a specific preference model (blue/red lines) versus not influencing them (gray line) on reward learning. The large differences between the Regret-Question and Partial-Return-Question condition’s lines (blue and red) compared to the control condition’s line (gray) support Hypothesis 2.

---

> > ### Comment · Reviewer_gCy4 · 2024-11-24
> >
> > Thank you for your clarifications. Here are my responses:
> >
> > “PRIVILEGED EXPERIMENT Figure 6: The difference between the P_regret and P_reward lines in both subplots is very minimal and does not hold for all number of partitions.”
> >
> > Unless I am misunderstanding something, hypothesis 2 states that "presenting each segment's statistic leads to learning more aligned reward functions with the preference model corresponding to this statistic", the fact that the red and blue lines are so close to each other contradicts this because we would expect the red line to be much higher in the left plot and the blue line to be much higher in the right plot.
> >
> > “TRAINED EXPERIMENT Figure 8: The P_regret trained preferences always lead to better performance -- this seems to indicate that the P_regret training paradigm leads to better understanding of the ground-truth reward function more than it supports Hypothesis 2.”
> > The problem is the same as above where we should expect the red line to be higher than the blue line in the left plot.
> >
> > “QUESTION EXPERIMENT The P_regret question does not seem to me to focus subjects on a segment's deviation from optimality. Perhaps something like "Which path is farthest from optimal?" would have been better Figure 10: The difference between the P_regret and P_reward lines in both subplots is very minimal and does not hold for all number of partitions.”
> >
> > The concern with Figure 10 is the same as both above.
> >
> > Regarding the wording of the question, I maintain that I believe that it does not reflect deviation from optimality as the authors claim.

---

> ### Author Response · Authors · 2024-11-25
>
> Thank you for your continued engagement with this work! Below are our responses
>
> "Unless I am misunderstanding something, hypothesis 2 states that "presenting each segment's statistic leads to learning more aligned reward functions with the preference model corresponding to this statistic", the fact that the red and blue lines are so close to each other contradicts this because we would expect the red line to be much higher in the left plot and the blue line to be much higher in the right plot."
>
> Your misunderstanding lies in the fact that hypothesis 2 can be restated as "presenting each segment's statistic leads to learning more aligned reward functions with the preference model corresponding to this statistic **compared to not presenting such information**". Therefore, to evaluate hypothesis 2, you should look at the difference between the red/blue lines and the gray line.
>
> "“TRAINED EXPERIMENT Figure 8: The P_regret trained preferences always lead to better performance -- this seems to indicate that the P_regret training paradigm leads to better understanding of the ground-truth reward function more than it supports Hypothesis 2.” The problem is the same as above where we should expect the red line to be higher than the blue line in the left plot."
>
> We are only interested in comparing a condition where we influence human subjects towards a specific preference model (red/blue lines) with the control condition (gray line). We should expect the red line to be higher than the gray line in the left plot but we do not see that. Hence, in Section 5.2, we claim that Hypothesis 2 is only partially supported; it is supported by the graph on the right not on the left.
>
> We hope that we have clarified how to analyze Figures 6,8, and 10 with respect to Hypothesis 2, and that this sheds light on the Question expirement.
>
> "Regarding the wording of the question, I maintain that I believe that it does not reflect deviation from optimality as the authors claim."
>
> Perhaps you are correct; we do not necessarily dispute your claim and further investigating the wording of this question is an interesting direction for future work. Regardless, in this paper, we do not see statistically significant results from this 1 condition in the Question expirement. Therefore we are unable to (and do not) claim that this specific question influences human subjects towards the regret preference model. Again, this may be because the wording of the question is not sufficient as you claim, but such an investigation is out of scope for this paper.

---

> > ### Author Response · Authors · 2024-11-26
> >
> > We note that, given your feedback, we have added the following to the caption of Figure 6:
> >
> > > To visually test for Hypothesis 2, observe the gap between the colored (i.e., red and blue) lines and the gray line.

---

> > > ### Comment · Reviewer_gCy4 · 2024-11-29
> > >
> > > Thank you for your response, however I disagree that the plots support hypothesis 2 and therefore maintain my score.

---

### Official Review · Reviewer_jTpL · 2024-11-02

**Soundness:** 3
**Presentation:** 4
**Contribution:** 3
**Rating:** 6
**Confidence:** 4

**Summary:**

This paper looks to test the idea that we may be able to influence people to behave more closely to target preference models. The motivation for this is twofold. (1) There remains a modelling gap in modelling human preferences, which makes learning underlying reward functions from those preferences harder and less accurate. This gap could be closed by improving said models, but — as this paper points out — it could equally be closed by making people behave more like the models we have now. (2) Even if perfect models of human preference existed, humans could still be influenced to behave like models on which inference is more tractable.

To this end, three interventions are proposed that could achieve this. Through three user studies on a grid world RL problem, it is tested for each intervention whether the intervention indeed influences people to give preferences that more closely align with a target preference model, and whether inference with the target model leads to more aligned inferred reward functions. The interventions are tested with two target preference models: one that assumes people prefer minimal regret behavior, and one that assumes people prefer maximal partial return behavior. The interventions are:
- “Privileged”, where the intervention consists of showing participants the partial return or regret of the pair of trajectories between which they must choose. This requires knowledge of the true reward function.
- “Trained”, where participants are trained through instruction to choose segments that maximise partial return or partial regret, before their preferences are elicited.
- “Question”, where how the text instruction that tells participants how to choose between the two trajectories is changed to better reflect partial return or regret comparisons.

Experimentally, the papers generally finds significant effects under all three interventions in terms of making people behave more like the target model, and in inferring a more aligned reward function.

**Strengths:**

I like the angle this paper takes very much. Human preference in settings such as the one considered here is very much an open problem. We only have limited understanding of how people value the options presented to them (see Knox et al. 2022) and limited understanding of how they would choose a most preferred option based on these values. With the (renewed) importance of preference learning, this is an important problem that remains unsolved in ML and even in cognitive science. The proposed solution of making people behave more like the simple preference models we have now is quite creative, and could go some way towards closing this gap, even if we must still recognise that people have inherent cognitive limitations that cannot be trained/influenced out.

The experiments are well-documented and the user study as described has been carried out to a level that is commensurate with what is expected at a top-level ML conference. The writing is clear and well-structured and the paper is easy to follow.

**Weaknesses:**

Primarily, I’m worried that the experiments carried out in the paper do not fully support the main claim of the paper. The stated goal is to test whether we can influence humans to follow a specific preference model. To test this, the authors check whether influencing someone towards a target preference model makes them behave more like that preference model compared to when they are not influenced (control condition). However, it is not established whether the interventions actually influence people to behave more like the target preference model as opposed to other preference models, or whether the intervention has solely influenced them to make better comparisons.

There seems to be some evidence of this in Figures 6 and 10, which show that  influencing participants to follow either one of the preference models leads to better reward inference with both, not only the targeted one. Furthermore, Figures 8 and 27, show that training people to follow a regret preference model is most effective for inferring their reward function, be that done with regret or partial return preference models. It seems to me then that although these interventions have a positive effect, this effect is not most specific to the target preference model.

In addition, the practical relevance of the proposed interventions is limited. Interventions in the “privileged” experiment are not practical because displaying true regret or true partial return is impossible without access to the unknown reward function (which the authors acknowledge). The intervention in the “trained” experiment is to train people to follow regret or partial return preferences. However, within the experiment presented here that training relies heavily on the true reward function (see questions). It is not clear how such training would work if the true reward function is not known. I see no practicality issues with the “question” experiment, but the effect size here is the most limited out of the three.

**Questions:**

Is training practical when the correct reward function is not known? In the experiment presented here, the true reward function was used in training, e.g. participants were corrected when they made wrong choices by saying something along the lines of “wrong, both items have equal score so far, but right option has higher possible score increase”. How would this training be implemented if the true reward function was not known?

For all three experiments, what was the mean Cross-Entropy Loss for the opposite preference model? For example, after a participant was trained to follow a partial return preference model, what was the likelihood of a regret preference model on that participant’s elicited answers?

For the question experiment, would participants have understood the difference between the question “which path has better immediate outcomes” and “which path reflects better decision-making”? The second questions seems ambiguous to me, considering the decision-making could be good in both the short or long term.

Section D.5 in the appendix suggests that people generally are better aligned with regret-based preference models. This is also in line with the work of Knox et al. 2022. Could this natural alignment towards regret-based preference have made training people to follow partial return preference less effective?

---

> ### Author Response · Authors · 2024-11-16
>
> We thank you for your review, and for your clear understanding of our work. We hope that our response to all reviewers has adequately addressed your concern that the experiments in our paper may not fully support the main claim. To summarize, additional results in Appendix H show that for all experiments, each condition’s dataset is more likely under the target preference model than the other condition’s datasets under the same preference model. This provides empirical evidence that the proposed interventions do in fact influence humans towards the target preference model, more so than towards the alternative preference model. To address your additional concerns around this point:
>
> “...Figures 6 and 10...show that influencing participants to follow either one of the preference models leads to better reward inference with both, not only the targeted one. “
>
> We agree with your observation here, and it highlights the general benefits of our proposed approach; influencing humans towards a specific preference model generally results in better reward inference with either preference model. This result is to be expected; partial return is a component of regret (see Eq. 7), and so influencing human preferences towards the partial return model may also influence their preferences towards the regret model and vice versa. We do not take this as evidence against any of our hypotheses.
>
> “Figures 8 and 27, show that training people to follow a regret preference model is most effective for inferring their reward function, be that done with regret or partial return preference models. “
>
> We also agree with your observation here, which also highlights a strength of our proposed approach. Training humans to follow the regret preference model benefits reward learning with either preference model. These results are in line with prior work, which found that learning from preferences generated by the regret preference model induces more aligned reward functions (see Knox et al., 2022) and that preferences generated by the regret preference model—when learned from under the assumption that instead they were generated by the partial return preference model—results in a highly shaped reward function in a variety of conditions (see Knox et al., 2024).
>
> We have addressed your concern around the practicality of our proposed approach in the response posted for all reviewers and in Appendix A. Regarding your question, “Is training practical when the correct reward function is not known?”, we propose exploring whether we can train human subjects to follow a specific preference model in a simple domain, such as the grid-world used in our work, with the goal of influencing their preferences in a more complex, real-world domain where the ground-truth reward function remains unknown. We leave this direction to future work outlined in Appendix A; this paper serves as a strong motivation for such a direction.
>
> To answer your remaining questions:
>
> “For all three experiments, what was the mean Cross-Entropy Loss for the opposite preference model?”
>
> We present these results in Appendix Figures 20-22.
>
> “For the question experiment, would participants have understood the difference between the question “which path has better immediate outcomes” and “which path reflects better decision-making”? The second questions seems ambiguous to me, considering the decision-making could be good in both the short or long term.”
>
> You make an excellent point. With regards to the second question, the wording was chosen to be somewhat ambiguous by design; we leave it up to the human subject to decide what constitutes “better decision-making”, including whether that means taking into account short-term or long-term decision-making. With regards to whether participants “would have understood the difference between the questions”, our empirical results do not measure exact understanding but do show that subjects were affected by the wording of the question, hinting at a strong level of understanding. We find that we can influence subjects towards the partial return model with statistical significance by only changing the wording of the question (see Section 5.3), indicating that even though the change in wording may appear subtle, it can have significant effects on how humans label preferences.
>
> “Section D.5 in the appendix suggests that people generally are better aligned with regret-based preference models. This is also in line with the work of Knox et al. 2022. Could this natural alignment towards regret-based preference have made training people to follow partial return preference less effective?”
>
> We find that, across all experiments, we can influence humans to follow the partial return preference model with statistical significance. Therefore, despite humans being generally more aligned with the regret preference model, all our proposed interventions prove effective at aligning humans with the partial return model.

---

> > ### Comment · Reviewer_jTpL · 2024-11-22
> >
> > Thank you for your detailed comments. These have cleared up some of the questions I had.
> >
> > Taking into account your comments and the new results in appendix H, I do agree that the experimental evidence in the paper supports its main claim.
> >
> > My concern regarding the practicality of the proposed interventions remains unchanged. The directions for future work proposed in appendix A are interesting, but as far as the current paper stands, only the question condition appears practical. I do agree that the results for the trained condition "suggest the potential efficacy of training humans" (as you point out in the global response).
> >
> > However, on balance, my view of the paper is now more positive. I think the research direction itself is promising. The experiments show significant effects in influencing humans towards specific preference models under various interventions, of which at minimum one is practical. Therefore I have raised my score.

---

### Official Review · Reviewer_T9uU · 2024-11-04

**Soundness:** 2
**Presentation:** 3
**Contribution:** 2
**Rating:** 5
**Confidence:** 4

**Summary:**

This paper focuses on reducing the gap between actual human behavior and the human models used in preference learning. While other work focuses on developing more complex human models, this submission focuses on changing human behavior to better conform to simple human models. The authors explore how to do this through a human study, focusing on three interventions. In the first intervention, they show annotators an explicit return or regret value for each partial trajectory in a comparison pair. In the second intervention, they train annotators to better estimate return or regret but then do not provide the actual return/regret values during annotation. In the third intervention, they simply change the language used to prompt the annotators. They find that the first two interventions lead to statistically significantly better agreement between human behavior and human models. The third also seems to increase agreement but is not statistically significant given the sample size.

**Strengths:**

The paper is generally quite well motivated and written. The math and experiments are clearly described. I am relatively familiar with this area and do not know of much other work which has focused on making human behavior more similar to human models rather than human models more similar to human behavior, so I think it is a novel direction. The experiments seem to be carefully designed and executed, and it seems helpful to the preference learning community to have well-grounded data on how well human models fit annotations.

**Weaknesses:**

The main weakness I see in the paper is that the first two proposed interventions do not actually seem to be applicable in practice. While the authors concede that the "privileged" setting is impractical, the second setting—"trained"—also seems to be impossible in practice. This is because the trained setting requires teaching people how to evaluate a particular aggregation of reward, which relies knowing the reward function in the first place; however, the entire point of preference learning is that the reward function is *unknown*. The third setting, "question," seems to be most applicable, but has the least convincing evidence.

In general, the authors argue that their interventions are focused on "training subjects to follow a preference model." However, it seems that to a large extent the authors are training subjects to follow a particular reward function—the privileged and trained experiments are both focused on helping annotators better estimate rewards. It would be more convincing that the authors are helping subjects "follow a preference model" if, for example, the subjects were trained to estimate return/regret with one reward function and tested with a different reward function. This would help disentangle the effects of simply learning more about the reward function and actually learning to follow one of the preference models.

Smaller issues/suggestions:
 * It would be helpful to include error bars in figures 5, 7, and 9.
 * There are a couple of improvements to the LaTeX math notation that could be made. For example, use `\text{loss}` instead of `loss` and `\text{regret}` instead of `regret`; use `\log` instead of `log`; and use `\mid` instead of `|`; use `\left` and `\right` with parentheses to make it clearer which parentheses match.
 * Line 345: "signficant" -> "significant"
 * Line 471: this sentence is confusing because it's referring to the experiments from Section 5.2. I think maybe it was meant to be replaced with links to the experiments in Section 5.3.

**Questions:**

* How could one actually implement the "trained" intervention in practice for an unknown reward function?
* How can we disentangle how much of the improvement in matching the human models is due to just learning the reward function better versus following the preference model?

---

> ### Author Response · Authors · 2024-11-16
>
> Thank you for the thoughtful review! We have addressed your concern regarding the practicality of our proposed approach in the response posted for all reviewers and in Appendix A. With regards to your question “How could one actually implement the "trained" intervention in practice for an unknown reward function?”, we propose investigating whether we can now train human subjects to follow a specific preference model in a simple domain, such as the grid-world domain we used in this work, with the goal of influencing their preferences in a more complex, real-world domain where the ground-truth reward function is unknown. We leave this direction to future work—outlined in Appendix A—where this paper serves as a strong motivation for such a direction.
>
> Additionally, we have also addressed your concern about disentangling the effects of simply teaching subjects more about the ground truth reward function rather than the chosen preference model; details in the Appendix of our paper support our hypothesis that this effect is likely relatively minimal, although future work will certainly focus on disentangling these factors more rigorously. Given your feedback, Appendix H now explicitly addresses your concern. The response posted for all reviewers and Appendix H therefore provides a partial answer to your question “How can we disentangle how much of the improvement in matching the human models is due to just learning the reward function better versus following the preference model?” We plan to further address this question in future work—now outlined in Appendix A—where we will elicit human preferences from a domain with a ground-truth reward function that is different from the one used for teaching subjects about a specific preference model.
>
> In your paper summary, you state “the third [intervention] also seems to increase agreement but is not statistically significant given the sample size.” We point out that the third intervention—the Question experiment—does yield statistically significant results for the condition where we influence human preferences towards the partial return model (see Section 5.3).
>
> Thank you for your attention to detail regarding the typos - we have corrected these in the revised manuscript. Regarding the addition of error bars to Figures 5, 7, and 9: We appreciate this suggestion but note that the choice of error bars requires careful consideration. The likelihoods of our given dataset are not normally distributed, hence the non-parametric statistical tests. Therefore, traditional error bars showing standard error or standard deviation would be misleading since these metrics assume normality. We note that the reported significance test results provide much of the value of error bars, though we plan to include them to maximize readability. We would like to spend more time thinking about what error bar metrics would be most appropriate to ensure they accurately reflect the statistical properties of our data. We are open to suggestions you may have as well!

---

> > ### Comment · Reviewer_T9uU · 2024-11-25
> > **Response to rebuttal**
> >
> > Thank you for the response to my concerns. I'm now more convinced that the experiments show that the approach is effective beyond just teaching annotators more about the reward function. I have a few more pieces of feedback:
> >
> >  * "Human models" can refer to many aspects of predicting/modeling human behavior, including irrationality (e.g., Boltzmann rationality, bounded rationality), risk aversion (e.g., expected utility theory, prospect theory), and others. It would be be helpful to be more clear that this paper focuses on how annotators temporally aggregate reward, which is just one aspect of a human model. It would also be good to acknowledge that there are many other ingredients in human modeling. The current framing of the title, abstract, and introduction suggests that the approach is generally applicable to other aspects of human modeling, but it's not clear if the approach generalizes beyond the particular comparison of human models the authors consider.
> >
> >  * Given that both reviewer jTpL and were concerned about the practicality of applying the given interventions when the reward function is unknown, I think that it would be best if some of the material in Appendix A could be moved to the main text. Furthermore, the results in Figures 20-22 are particularly important for showing that the interventions are not simply teaching annotators more about the reward function. In particular, in Figure 21, the fact that training annotators for one model actually makes them have higher cross entropy under the other model suggests that the intervention is creating the desired effect. Thus, I feel like it would be helpful to include in Figures 5, 7, and 9 an additional bar for subjects intervened-on towards the opposite model. It could also be helpful to proactively mention this concern in the main text and how these results show that the effect is not coming from learning more about the reward function.
> >
> > I have raised my score for now. Besides the suggestions above, a particularly valuable addition to the paper would be to show that annotators trained to follow a human model for one reward function are also influenced towards that human model for other, previously unseen reward functions. I know that this is not realistic for this review cycle but it could be a way to eventually improve the paper's impact by demonstrating the practicality of the approach.

---

> > > ### Author Response · Authors · 2024-11-26
> > >
> > > We greatly appreciate your continued engagement and valuable feedback, which have strengthened our work!
> > >
> > > To your first point of feedback, you make an excellent point; In section 3.2 we add the following:
> > >
> > > > We focus on two preference models that differ solely in the segment statistic they use to evaluate the desirability of a trajectory segment. We do not address other aspects of modeling human preferences, such as different assumptions about human irrationality, risk aversion, or uncertainty. However, the interventions proposed in this paper, which aim to influence humans towards a chosen preference model, may be generalizable to influencing humans to conform to preference models that incorporate these additional factors.
> > >
> > > Your feedback highlights some interesting questions for the RLHF community and potentially impactful directions for future work; namely, can humans be influenced toward other preference models that make different assumptions about human rationality and risk averseness?
> > >
> > > To your second point of feedback, we have added the following paragraph to our conclusion, building on Appendix A and addressing your’s and reviewer jTpL’s concerns:
> > >
> > > > Our findings suggest that human training and preference elicitation interfaces should be viewed as essential tools for improving alignment in RLHF. Appendix A highlights the potential practicality of our approach; the Trained and Question interventions demonstrate important potential for real-world application. Notably, the Question experiment offers a viable path forward by influencing human preferences towards a specific preference model without requiring knowledge of the ground-truth reward function, while the Trained experiment establishes a foundation for extending this methodology to real-world domains.
> > >
> > > Finally, to address your concern about the possible entangled effects of learning more about the ground-truth reward function versus the chosen preference model, we have added the following in section 5.1
> > >
> > > > Additionally, Appendix H provides an explanation of how we disentangle the effects of learning more about the ground-truth reward function versus the target preference model, addressing concerns that the proposed interventions may merely teach human subjects more about the ground-truth reward function rather than the target preference model.
> > >
> > > We extensively discussed adding an additional bar for subjects intervened-on towards the opposite preference model in Figures 5, 7, and 9. Our resulting consensus was that we do not want readers to make comparisons between intervention conditions—which we do not make claims about and which we have not extensively investigated—we only want readers to make comparisons between an intervention condition and the control condition. As such, we do not want to invoke such comparisons by plotting that data, particularly when it is not needed to analyze hypothesis 1. Instead, we hope that the sentence we added above points readers to the relevant plots, particularly if they share your concerns.

---

### Author Response · Authors · 2024-11-16
**Response to all reviewers**

We thank the reviewers for their thoughtful feedback and for recognizing the novelty of our proposed research direction. Here we address the concerns shared by several reviewers, followed by reviewer-specific responses below.

Several reviewers (T9uU, jTpL, 6yvA) raised concerns about the practical applicability of our approach, particularly regarding the Trained experiment's reliance on the ground-truth reward function in a grid-world domain. While we acknowledge this limitation, our work makes several important contributions that outweigh this concern:

(1) Novel Research Direction: To our knowledge, this is the first work exploring how to systematically influence human preference expression to better match RLHF algorithmic assumptions. This represents a fundamentally new and potentially impactful approach to improving model alignment.

(2) Proof of Concept: All experiments demonstrate that human preference expression can indeed be systematically influenced to better match specific preference models. While we used the ground-truth reward function for evaluation purposes, this establishes the core feasibility of our approach.

(3) Clear Path to Generalization: The Trained experiment lays the groundwork for extensions to more complex domains and scenarios where the ground-truth reward function is unknown or is known for the purposes of experimental evaluation—as in our experiments—but differs from any reward function(s) used during training. The significant effects observed in the Trained experiment suggest the potential efficacy of training humans to follow a chosen preference model, which is an important insight for the broader RLHF community.

(4) Practical Implementation: The Question experiment already demonstrates a viable path forward, showing significant results for one condition in which the intervention does not rely upon any knowledge of the ground-truth reward function. This provides immediate practical value while suggesting promising directions for future work.

We have added a summary of these arguments in Appendix A.

Reviewer T9uU raised the concern that we may be training humans to better understand the ground truth reward function, rather than to follow a specific preference model. We acknowledge the possibility of entangled effects relating to learning more about the reward function rather than a specific preference model. However, we expect that in our experiments such effects are relatively minimal. Firstly, all human subjects already had a good understanding of the ground-truth reward function; we employed a comprehension test to filter out subjects who did not (see Appendix C.2). Further, in Appendix Figures 20-22, we see that for all experiments the loss over a condition’s dataset is lower under the target preference model than all other conditions datasets under the same preference model. Had one condition trivially resulted in subjects better understanding the ground-truth reward function rather than the target preference model, we would expect to see that condition’s dataset induce the lowest loss under either preference model. Appendix Figures 20-22, illustrate this is not the case. We have added this explanation to Appendix H of the paper. To further address this concern, our future work will focus on training human subjects to follow a specific preference model with one reward function, after which we will elicit human preferences in another domain with a different ground truth reward function. We have added this direction of future work to Appendix A.

Reviewers jTpL and gCy4 raised the concern that the proposed interventions may influence humans towards another preference model rather than the target preference model, and that while the interventions had a positive effect, “this effect is not most specific to the target preference model”. Our empirical evidence suggests otherwise: Appendix Figures 20-22 present results extending those in Figures 5,7, and 9, illustrating that for all experiments, influencing human subjects towards a specific preference model resulted in that corresponding dataset being more likely under the target preference model than all other datasets. Therefore, the effect of each intervention across all conditions is most strong for the target preference model. Naturally, however, aligning human preferences closer with regret may also align them closer with partial return and vice versa. This does not undermine our supported hypothesis that, with each intervention, we are able to influence human preferences towards a specific preference model.

---

### Meta-Review · Area_Chair_VDfb · 2024-12-21

**Metareview:**

This paper addresses RLHF, specifically by changing human behaviour to better look like target preference models. This is a new direction of research and is interesting, which reviewers agreed. Reviewers also found the paper well-writen and clear (in terms of arguments and structure), and the experiments well-documented.

The main weakness in the paper was also agreed across most reviewers, that the experiments are not practical (beyond the grid-world setting / knowing true rewards). Most reviewers agree that this is a big enough limitation to overcome the strengths of the paper, and my own reading of the paper is the same. The authors wrote some future directions in Appendix A, which are interesting, but not enough in my assessment.

Reviewer gCy4 disagrees that the plots support Hypothesis 2, and this is the main reason for their score. I am discounting this negative comment because I agree with the authors on this point. Reviewer jTpL also changed their mind during rebuttal to say that the experiments do support the overall paper claims.

**Additional Comments On Reviewer Discussion:**

Please see overall metareview. A major concern is that the experiments (beyond the question experiment, which has the least convincing evidence) are not practical. Tackling this beyond a few paragraphs in Appendix A will improve this work further in a future version.

Reviewer gCy4 disagrees that the plots support Hypothesis 2, and this is the main reason for their score. I am discounting this negative comment because I agree with the authors on this point. Reviewer jTpL also changed their mind during rebuttal to say that the experiments do support the overall paper claims.

The rebuttal also addressed various misunderstandings about understanding figures and experimental results, which the reviewers mostly appreciated (such as Reviewer jTpL), and improve the readability of the paper further.

---

### Decision · Program_Chairs · 2025-01-22

Reject